# Leaf Proteomic Analysis in Seedlings of Two Maize Landraces with Different Tolerance to Boron Toxicity

**DOI:** 10.3390/plants12122322

**Published:** 2023-06-15

**Authors:** Betty Maribel Mamani-Huarcaya, María Teresa Navarro-Gochicoa, María Begoña Herrera-Rodríguez, Juan José Camacho-Cristóbal, Carlos Juan Ceacero, Óscar Fernández Cutire, Agustín González-Fontes, Jesús Rexach

**Affiliations:** 1Departamento de Fisiología, Anatomía y Biología Celular, Universidad Pablo de Olavide, E-41013 Sevilla, Spain; bmamanih@unjbg.edu.pe (B.M.M.-H.); mtnavgoc@upo.es (M.T.N.-G.); mbherrod@upo.es (M.B.H.-R.); jjcamcri@upo.es (J.J.C.-C.); cjcearui@upo.es (C.J.C.); agonfon@upo.es (A.G.-F.); 2Laboratorio de Biotecnología Vegetal, Escuela de Agronomía, Facultad Ciencias Agropecuarias, Universidad Nacional Jorge Basadre Grohmann, Tacna 23000, Peru; 3Departamento de Agronomía, Facultad Ciencias Agropecuarias, Universidad Nacional Jorge Basadre Grohmann, Tacna 23000, Peru; ofernandezc@unjbg.edu.pe

**Keywords:** boron toxicity, proteomic analysis, maize landrace, *Zea mays*

## Abstract

Boron (B) toxicity is an important stressor that negatively affects maize yield and the quality of the produce. The excessive B content in agricultural lands is a growing problem due to the increase in arid and semi-arid areas because of climate change. Recently, two Peruvian maize landraces, Sama and Pachía, were physiologically characterized based on their tolerance to B toxicity, the former being more tolerant to B excess than Pachía. However, many aspects regarding the molecular mechanisms of these two maize landraces against B toxicity are still unknown. In this study, a leaf proteomic analysis of Sama and Pachía was performed. Out of a total of 2793 proteins identified, only 303 proteins were differentially accumulated. Functional analysis indicated that many of these proteins are involved in transcription and translation processes, amino acid metabolism, photosynthesis, carbohydrate metabolism, protein degradation, and protein stabilization and folding. Compared to Sama, Pachía had a higher number of differentially expressed proteins related to protein degradation, and transcription and translation processes under B toxicity conditions, which might reflect the greater protein damage caused by B toxicity in Pachía. Our results suggest that the higher tolerance to B toxicity of Sama can be attributed to more stable photosynthesis, which can prevent damage caused by stromal over-reduction under this stress condition.

## 1. Introduction

Boron (B) is an essential element for plants, and is well known for its structural role in both cell walls and membranes [1,2,3,4,5]. In fact, B establishes diester bonds between the apiose residues of two rhamnogalacturonan-II (RGII) molecules, forming RGII-B complexes that stabilize the pectin network of the cell wall [6,7,8]. Furthermore, B contributes to the preservation of plasmalemma integrity and function [9], likely through the formation of B complexes with membrane components that contain cis-diol groups [10,11]. Thereby, B forms complexes with major constituents of membrane lipid rafts, such as glycosyl inositol phosphoryl ceramides (GIPCs) [12]. Moreover, B participates in the formation of GIPCs-B-RGII complexes, which connect the plasmalemma to the cell wall [13]. Besides these structural roles, B is also involved in plant development, participating in root and shoot elongation, pollen-tube growth, flowering, and fruiting [14,15,16]. In addition, B has been reported to participate in several physiological processes, such as photosynthesis; nucleic acid synthesis; phenolic, nitrogen and polyamine metabolism; protein stabilization and biosynthesis; and gene expression, among others [16,17,18,19,20,21,22].

Since B is a micronutrient, the range between its deficient, optimal, and toxic concentrations for plants is very narrow [23]. Therefore, it is common to find soils with inadequate B content for optimal plant development. Soils with high B content predominantly occur in arid and semi-arid countries, where this micronutrient accumulates in the topsoil, mainly owing to high evapotranspiration and low leaching caused by low rainfall, a situation that is often aggravated by irrigation with B-enriched water [22,23]. Additionally, excess B is also found on land close to coastal areas due to the hydraulic connection between their coastal aquifers and seawater [24], or in regions with recurrent geothermal activities [2]. Climate change is another factor that contributes to the B increase in soils. Increasing temperatures and decreasing rainfall are predicted in the coming years, which will lead to an increase in agricultural areas with excessive B levels [3,25]. 

Excessive B content in soils causes adverse effects, such as chlorosis and necrosis in leaves, damage to stems and buds, and misshapen fruits [17,22]. Furthermore, an excess of B induces DNA damage, the inhibition of protein folding, the impairment of protein functions and activities, and alterations to photosynthesis and to nitrogen and carbon metabolism, among other processes [2,22,26]. In fact, several photosynthetic parameters, such as CO_2_ assimilation (P_N_), photosynthetic electron transport rate (ETR), maximum quantum yield of chlorophyll fluorescence (Fv/Fm), and CO_2_ use efficiency decreased under B toxicity conditions [22,27]. Because of the aforementioned effects of B toxicity in plant physiology, elevated B content in agricultural lands reduce crop growth, yield, and quality [22,28]. In fact, a noteworthy decrease in the yield of several main crops subjected to B toxicity has been reported [28]. Despite the large number of effects caused by B toxicity in plants, it is not well known how B produces these alterations. However, it has been suggested that the ability of B to form bonds with molecules containing mono-, di-, and poly-hydroxyl groups could be the chemical basis by which B toxicity triggers morphophysiological alterations [29].

Maize is an important crop that provides approximately half of the calories consumed worldwide; in addition, it is one of the principal genetic model plants for crop improvement and food security [30,31,32]. However, maize production is seriously constrained by abiotic and biotic stresses [33]. In particular, B toxicity causes a decrease in maize production, as well as in other cereals [28,34,35]. Therefore, the search for and molecular characterization of new maize varieties with improved tolerance to B toxicity has become an interesting research topic. In a recent work, two Peruvian maize landraces (Pachía and Sama) were tested for tolerance to high B. The Sama landrace had greater tolerance to B excess than Pachía [27]. In this work, a comparative proteomic characterization of these two maize landraces with different tolerance to B toxicity was performed to improve our molecular knowledge about which proteins are involved in B toxicity tolerance. To better visualize the differences between the control and B toxicity treatments, proteomic analyses were carried out on leaves, since this tissue has higher B content than the roots, and the differences between the two treatments should be greater.

## 2. Results

A total of 2793 proteins were identified in at least one of the biological replicates of a landrace (Sama or Pachía) and a B treatment analyzed (Appendix A). In addition, the number of proteins detected in both Pachía and Sama in each of the B treatments studied was similar at close to 1100 proteins (Table 1). 

Appendix A shows the dataset of the identified proteins, indicating their gene ontology (GO), biological processes (GOBP), GO molecular functions (GOMF), and GO cellular compartments (GOCC), and Appendix A summarizes the statistical analysis and fold changes in the proteins. To study the differentially accumulated proteins in Pachía and Sama in both B treatments, four comparison groups were established: (1) Sama and Pachía seedlings subjected to the control B condition (S0.05/P0.05), (2) Sama and Pachía treated with 10 mM B (S10/P10), (3) Sama subjected to 10 mM and 0.05 mM B (S10/S0.05), and (4) Pachía treated with 10 mM and 0.05 mM B (P10/P0.05). A total of 303 proteins had statistically significant differential expression (*p* ≤ 0.05) in the above groups (Appendix A). The S0.05/P0.05 and S10/P10 groups contained those proteins that were differentially expressed between Sama and Pachía in 0.05 mM or 10 mM B, respectively. In media with 0.05 mM B, more proteins were up- and down-accumulated between Sama and Pachía than in 10 mM B (Figure 1 and Table 1). In addition, the S10/S0.05 and P10/P0.05 comparison groups included proteins that were differentially expressed in response to B toxicity in Sama or Pachía, respectively. Pachía had a higher number of proteins induced and repressed by B toxicity than Sama; thus, 98 proteins were overexpressed in Pachía in 10 mM B, while only 38 were overexpressed in Sama, and 51 proteins were underexpressed in Pachía under B toxicity versus 28 in Sama (Figure 1).

### 2.1. Classification into Several Functional Categories of Differentially Accumulated Proteins in Both Maize Landraces and B Treatments

All significant differentially expressed proteins in the four comparison groups described above were functionally classified into 26 categories using several databases (Appendix A). The functional categories that included the largest number of differentially accumulated proteins were transcription and translation processes (57), photosynthesis (25), amino acid metabolism (24), protein degradation (23), carbohydrate metabolism (20), and protein stabilization and folding (18) (Figure 2 and Appendix A). These main categories together contained more than 50% of the total differentially expressed proteins.

### 2.2. Differentially Expressed Proteins in Sama and Pachía in Response to B Toxicity 

Considering that the major aim of this work was to analyze the changes induced by B toxicity in protein expression in Pachía and Sama, we will now focus on the proteins that were differentially expressed due to B toxicity in these landraces. Thus, 66 and 149 proteins were differentially expressed in response to B toxicity in Sama and Pachía, respectively (Table 1). The main functional categories containing the highest number of differentially expressed proteins under B toxicity in both Sama and Pachía were transcription and translation, photosynthesis, amino acid metabolism, protein degradation, protein stabilization and folding, and reactive oxygen species (ROS) (Figure 3 and Figure 4). Interestingly, most of the proteins belonging to the transcription and translation category were induced in response to B toxicity in both Sama and Pachía, with the number of differentially induced proteins being remarkably higher in Pachía (Figure 3 and Figure 4). However, almost all proteins included in the photosynthesis category were repressed in 10 mM B, with the number of down-accumulated proteins also being higher in Pachía than in Sama (Figure 4 and Appendix A). Regarding protein degradation, and protein stabilization and folding, most of the differentially expressed proteins in 10 mM B were found in Pachía, suggesting that B toxicity would alter the structure and folding of proteins in this landrace. In addition, many of the proteins in the ROS category were induced by B toxicity in both landraces (Figure 4 and Appendix A). Although the groups of carbon assimilation and metabolism, lipid metabolism, and respiration included a smaller number of proteins than those mentioned above, a larger number of differentially expressed proteins were found in Pachía under B toxicity (Figure 4 and Appendix A). Other interesting categories were cell death, cell division, cell wall, ribosome biogenesis, and RNA binding and processing, which, despite having a very small number of proteins regulated by B toxicity, had an interesting distribution in both landraces and B treatments. In fact, in the cell death and cell wall categories, only proteins whose expressions were induced by B toxicity were found in Pachía; however, the cell division, ribosome biogenesis, and RNA binding and processing categories also contained proteins with higher accumulation in 10 mM B in both landraces (Figure 4 and Appendix A).

A total of 18 proteins were commonly expressed (repressed or induced) in both landraces in response to B toxicity, with the amino acid metabolism and photosynthesis categories having the highest number of proteins (Table 2). All proteins of the amino acid metabolism group were up-accumulated under B toxicity conditions, with these inductions being slightly greater in Pachía than in Sama. Interestingly, however, all commonly expressed proteins from the photosynthesis category were repressed by B toxicity, with these repressions being remarkably higher in Pachía than in Sama (Table 2).

Appendix A list the most strongly differentially expressed proteins that were up- or down-regulated more than twofold by B toxicity in Pachía and Sama, respectively. In Pachía, 105 proteins had strong differential expression under B toxicity, while only 27 were found in Sama. Photosynthesis was the functional category containing the highest number of proteins whose expressions were strongly down-accumulated in response to B toxicity in both Pachía and Sama (Appendix A); however, interestingly, a minor number of both repressed and very strongly repressed (FC < 0.33) proteins were observed in Sama (Table 3, Table 4, Appendix A). Different subunits of the NDH complex (NDHS, B1, B2, J, and H) were strongly repressed by B toxicity in Pachía but not in Sama (Appendix A). In addition, only in Pachía did we detect proteins related to protein degradation processes whose expressions were mainly induced by B toxicity, suggesting that enhanced damage would be induced by 10 mM B in Pachía proteins (Table 3 and Appendix A). Furthermore, B toxicity markedly induced a larger number (15) of proteins in Pachía belonging to the transcription and translation category (Appendix A).

Table 5 shows the proteins that were strongly up- or down-accumulated when the protein expressions of Sama were compared to those of Pachía in media with 10 mM B. Sama had remarkable up-accumulation of four proteins involved in photosynthesis (ZmPIFI and OEE2-1), chlorophyll biosynthesis (ChlH1), and secondary metabolism (PAO1) (Table 5), with this last protein also being very strongly induced in response to B toxicity (Table 4). However, in Pachía, several proteins were detected to exhibit strong accumulation in 10 mM B when compared with Sama (shown in Table 5 as strongly down-accumulated proteins in Sama) highlighting, among them, histone H1 and ribosomal protein S7, which were very strongly induced by B toxicity (Table 3).

Finally, in both Pachía and Sama, proteins exclusively detected in one of these landraces were found; among them were Nfc103a and eIF3a, which were only identified in Pachía in 10 mM B (Table 6).

## 3. Discussion

The results of this work were obtained with seedlings of two maize landraces, Sama and Pachía, since at this stage of development, the plants are more sensitive to B excess. Increasing work is being performed to elucidate the molecular mechanisms that allow crop varieties to be tolerant to B toxicity [25,27,36,37,38]. The identification of proteins that contribute to B tolerance could be a useful tool to breed higher-yielding and higher-quality maize varieties. 

Although 2793 proteins were detected in this proteomic analysis, only 303 proteins were differentially accumulated (Appendix A), which were classified into 26 functional categories. Functional analysis indicated that pathways involved in the transcription and translation processes, amino acid metabolism, photosynthesis, carbohydrate metabolism, protein degradation, and protein stabilization and folding were highly enriched categories in both landraces (Figure 2). Remarkably, the expression levels of proteins related to these enriched processes were significantly different between Pachía and Sama.

### 3.1. Several Proteases and Translation-Related Proteins Allow Pachía to Survive in Media with B Excess

Pachía is a B-sensitive maize cultivar described by Mamani-Huarcaya et al. [27]. Interestingly, the highest number of differentially accumulated proteins (DAPs) was found in the comparison group P10/P0.05 (Figure 1) suggesting that the B toxicity damage caused in Pachía could be partially relieved by these proteins. A remarkable number of these DAPs included in the categories of protein degradation (11), and transcription and translation (15), were strongly overexpressed in Pachía (Appendix A). However, only five proteins of the transcription and translation group were markedly induced by 10 mM B in Sama (Appendix A). The B-sensitive *Citrus grandis* had a higher number of proteins involved in protein degradation and was also overexpressed under B toxicity conditions in comparison with B-tolerant *Citrus sinensis* [39]. These authors concluded that B toxicity caused greater protein damage and proteolysis in *C. grandis*. Therefore, the high number of protein degradation-related proteins that were overexpressed in Pachía in 10 mM B would suggest that B toxicity would cause greater damage in Pachía proteins than in those of Sama, leading to increased proteolysis in B-sensitive Pachía. Proteins related to protein degradation that were strongly overexpressed in Pachía included, among others, cysteine protease14 and four serine proteases (Appendix A). Proteases have been implicated in plant acclimation to abiotic stress, playing a major role in the degradation of damaged and misfolded proteins, thus contributing to cell survival. In fact, cysteine and serine proteases are involved in the degradation of misfolded proteins and protection against abiotic stresses [40,41,42,43]. Hence, these five proteases could have a main role in the degradation of damaged and misfolded proteins in Pachía under excess B, contributing to maintaining the correct conformation of Pachía proteins and, therefore, to the survival of this landrace under this stressful condition. In addition, a noteworthy number of proteins involved in transcription and translation processes were overexpressed at 10 mM B in Pachía (30, in contrast to only 9 in Sama) (Appendix A). Proteomic analysis performed with dehydration, salt, and temperature stresses in cereals also displayed alterations in the levels of translation-related proteins, such as initiation factors and the ribosome constituent proteins ([44] and references therein). Furthermore, it has been suggested that a B excess induces the inhibition of RNA-dependent processes, such as transcription and translation, owing to the ability of B to form complexes with ribose molecules [45]. In this regard, Tanaka et al. [46] have suggested that B or boric acid acts on the translation machinery, likely forming complexes with cis-diol groups of rRNA and tRNA. In addition, it has recently been proposed that high-B stress enhances ribosome frequency on stop codons, leading to global ribosome stalling [47]. Consequently, the high content of leaf-soluble B in Pachía seedlings subjected to 10 mM B, reported by Mamani-Huarcaya et al. [27], would generate an increased formation of B complexes with cis-diol groups of RNA, which would damage ribosomes, leading to a drop in protein synthesis, likely through global ribosome stalling. The strong overexpression of several ribosomal proteins would maintain the Pachía ribosome stability in B toxicity (Appendix A). These results are consistent with those reported for rice, where several ribosomal protein large subunit genes were upregulated under temperature stress; this suggests that their encoded proteins might be involved in stress amelioration, likely maintaining the proper functioning of ribosomes [44]. Interestingly, the eukaryotic translation initiation factor 3 subunit A (eIF3a) was exclusively detected in B toxicity in Pachía (Table 6). These factors are one of the most significant components involved in plant protein synthesis and, specifically, rice eIF3A has been proposed to play an important role in different stresses [48]. Therefore, eIF3a would also help to alleviate the drop in protein synthesis in Pachía. Thereby, Pachía would partly ameliorate injuries caused by B toxicity in protein synthesis and ribosomes by overexpressing a high number of transcription- and translation-related proteins, abolishing the non-viable reduction in transcription and translation processes.

### 3.2. Proteins That Can Confer More B Toxicity Tolerance to Sama

Polyamine oxidase 1 (PAO1) is an interesting protein that was clearly up-accumulated in Sama when compared to Pachía at 10 mM B, and was also very strongly induced in Sama by B toxicity (Table 4 and Table 5). This enzyme catalyzes the back-conversion of spermine (Spm) to spermidine (Spd), and Spd to putrescine (Put) [49]. Maize polyamines play a crucial role in the abiotic stress response [33]. In fact, it has been reported that Put protects the plant’s photosynthetic apparatus against several abiotic stresses [50]. Moreover, the conjugation of Put to PSII proteins may lead to the structural and functional stability of PSII [49,51]. Therefore, the over-accumulation of PAO1 in Sama plants subjected to B toxicity would generate an increase in Put levels that would protect their photosynthetic apparatus, resulting in the higher P_N_ observed in Sama under this stress, as described by Mamani-Huarcaya et al. [27]. This finding is consistent with the results reported for Karoon, a drought-tolerant maize cultivar. Pakdel et al. [49] proposed that higher expression of PAO genes and enzymatic polyamine oxidation activity can protect the photosynthetic apparatus of Karoon under water stress.

#### Lower Repression of Photosynthesis-Related Proteins Can Enhance the B Toxicity Tolerance of Sama

Photosynthesis is one of the essential physiological processes affected by B toxicity [2,22]. Photosynthetic efficiency can be achieved in Sama under B toxicity conditions by increasing the synthesis of photosynthetic pigments, since chlorophyll content is a major limiting component of photosynthetic efficiency [52]. Interestingly, Sama had a strong over-accumulation of magnesium-chelatase subunit H1 chloroplastic (ChlH1) at 10 mM B in comparison with Pachía (Table 5). ChlH binds to porphyrin and catalyzes the insertion of Mg^2+^ into protoporphyrin IX [53]. Accordingly, the over-accumulation of ChlH1 in Sama would explain its higher content of chlorophyll *a* in B toxicity and the higher P_N_ described by Mamani-Huarcaya et al. [27].

In this study, 25 proteins related to photosynthetic light reactions were differentially accumulated, with most of them involved in electron transport, light harvesting, and oxygen-evolving processes (Figure 2 and Appendix A). Pachía and Sama presented several photosynthesis-related proteins that were repressed by B toxicity when their expressions were compared with those of Pachía and Sama, respectively, in media with 0.05 mM B (Appendix A). However, the number of these DAPs was lower in Sama than in Pachía (11 versus 16, respectively; Appendix A); moreover, those proteins commonly down-accumulated in both landraces had a weaker decrease in Sama (Table 2). In addition, only two photosynthetic proteins were very strongly underexpressed (3-fold or more (corresponding to FC ≤ 0.33)) by B toxicity in Sama in contrast to ten proteins found in Pachía (Table 3 and Table 4). This decreased accumulation of photosynthesis related-proteins may cause lower photosynthetic performance in B toxicity-treated Pachía plants than in Sama plants, as described by Mamani-Huarcaya et al. [27]. Therefore, Sama can retain sufficient levels of photosynthesis-related proteins in 10 mM B, which allows it to maintain photosynthetic parameters at similar levels to those of the control conditions, as reported by Mamani-Huarcaya et al. [27]. Furthermore, three photosynthesis-related proteins were up-accumulated in Sama when their expressions were compared with those of Pachía in 10 mM B, namely, oxygen-evolving enhancer protein 2-1 chloroplastic (OEE2-1), post-illumination chlorophyll fluorescence increase (ZmPIFI), and NAD(P)H-quinone oxidoreductase subunit S chloroplastic (NDHS) (Table 5 and Appendix A). OEE2-1 is likely an extrinsic protein of the oxygen-evolving complex (OEC) (UniProt; https://www.uniprot.org/, accessed between 6 June 2022 and 24 January 2023). The OEC is stabilized and protected by extrinsic polypeptides [54]. The strong OEE2-1 over-accumulation in 10 mM B in Sama could facilitate the stability and protection of the OEC, leading to the higher photosynthetic electron transporter rate (ETR) observed in this landrace [27]. Regarding ZmPIFI, it is homologous to the PIFI protein of *Arabidopsis thaliana* (AtPIFI), an essential component of the NAD(P)H dehydrogenase (NDH) complex involved in chlororespiratory electron transport around PSI [55]. The *Atpifi* mutant had lower nonphotochemical quenching (NPQ) than the wild type under high light irradiances, suggesting that AtPIFI can protect plants from photooxidative stress triggered by excessive light [55]. Consequently, both ZmPIFI over-accumulation and the higher NPQ values that Sama showed in 10 mM B, unlike those from Pachía (Table 5; [27]), suggest that ZmPIFI is also a component of the maize NDH complex, playing a role in the oxidative photoprotection of this landrace under B toxicity conditions. Furthermore, unlike Sama, several subunits of the NDH complex were markedly repressed in Pachía by B toxicity (Appendix A). The NDH complex mediates cyclic electron transport around PSI, playing a crucial role in C_4_ photosynthesis [56,57]. NDH-mediated cycle electron transport (NDH-CET) performs two functions: (1) maintaining photosynthetic redox balance in electron transfer, avoiding stromal overreduction and functioning as a safety valve for excess electrons under stress, and (2) supplying ATP for efficient carbon assimilation, especially under stressful conditions [56,57,58,59]. The finding that none of the above components of the NDH complex was significantly repressed by B toxicity in Sama suggests that its NDH-CET can prevent stromal overreduction and protect against photooxidation. This fact would explain the high values of net photosynthetic CO_2_ assimilation (P_N_), maximum photochemical efficiency (Fv′/Fm′), and quantum yield efficiency of PSII electron transport (Φ_PSII_) reported in Sama at 10 mM B, which were similar to those of the control conditions [27]. Consistent with our data, Zhu et al. [59] have suggested that an increased abundance of NDH subunits in salt-stressed wheat would enhance NDH-CET, alleviating the accumulation of excess electrons and maintaining energy homeostasis. Moreover, the subunit S of the NDH complex was over-accumulated in Sama under B toxicity when compared to that in Pachía, likely leading to a higher amount of NDH complex, which would provide extra ATP to achieve better P_N_ and growth in this landrace in medium with 10 mM B as, in fact, was observed by Mamani-Huarcaya et al. [27]. In addition, a higher supply of ATP was obtained in Sama in comparison to Pachía under B toxicity due to a weaker decrease in the α- and β-chloroplastic subunits of ATP synthase in Sama (Table 2). Although B excess causes photosynthetic damage [2,22], plants have evolved mechanisms to repair these injuries that require a high amount of ATP from chloroplastic ATP synthase [60,61]. In Sama, B toxicity barely affected the photosynthetic parameters [27]. This finding indicates that this landrace possesses mechanisms to repair its photosynthetic machinery. It is likely that one of these mechanisms is to provide greater ATP availability, which can be achieved by maintaining sufficient levels of NDH and ATP synthase complexes to synthesize the amounts of ATP needed to repair its photosynthetic machinery and, therefore, to maintain its photosynthetic values at levels similar to those of control conditions.

Our data provide valuable information for future research to breed improved maize varieties against this abiotic stress. Nevertheless, it is known that results obtained under laboratory conditions are not always applicable to field conditions. Therefore, it would be interesting to evaluate the productivity of Sama and Pachía in soils with B excess. However, the strong evidence of higher photosynthesis-related protein content in Sama (Table 2 and Appendix A), which is supported by the higher values of photosynthetic parameters [27], suggests that field results may be very similar to those obtained under laboratory conditions in this work.

## 4. Materials and Methods

### 4.1. Plant Materials and Growth Conditions

Sama and Pachía, two Peruvian maize landraces from the Sama valley and the Pachía district (to the east of Tacna), were used in this study. Seeds were surface-sterilized as described by Mamani-Huarcaya et al. [27]. Afterwards, the seeds were placed in seedbeds filled with a perlite/vermiculite mixture (1/1, *v*/*v*) and watered with deionized H_2_O. After seven days, seedlings were transplanted to 30 L plastic containers with a nutrient solution (NS) that was identical to the one used by Mamani-Huarcaya et al. [27]. After two days of acclimation to hydroponic medium, the seedlings were divided into groups and transferred to fresh NS supplemented with 10 mM H_3_BO_3_ (B toxicity conditions) or 0.05 mM H_3_BO_3_ (control conditions). This medium was aerated by air pumps and renewed twice a week. The seedlings were germinated and grown hydroponically in a growth chamber under a 12 h light/12 h dark regime (215 µmol m^–2^ s^–1^ of photosynthetically active radiation at plant height), at 22 °C and 50% relative humidity. The plants were randomly harvested 10 days after the onset of the B treatments, and their leaves were quickly separated with a scalpel, frozen in liquid nitrogen and stored at –80 °C until further analysis.

### 4.2. Protein Extraction and Digestion

Maize leaves (200–250 mg fresh weight) from four separate seedlings per condition (B treatment and maize landrace) were ground to a fine powder in a mortar precooled with liquid nitrogen. Proteins were extracted with trichloroacetic acid (TCA)/acetone-phenol [62], solubilized in a solution containing 7 M urea, 2 M thiourea, and 2% (*w*/*v*) CHAPS (3 [(3-cholamidopropyl) dimethylammonium]-1-propanesulfonate), and quantified via the Bradford method using bovine serum albumin (BSA) as a standard [63].

The cleaning of maize protein extract, protein digestion, and mass spectrometry determinations were carried out at the Proteomics Facility for Research Support Central Service (SCAI) of the University of Córdoba (Spain) as follows.

Biological quadruplicate samples were separated and cleaned as described. Leaf protein extracts (50 µg of BSA protein equivalents per sample) were electrophoretically pre-concentrated in a centimeter band of 10% (*w*/*v*) SDS-PAGE gel. Protein bands were excised from the gels and, afterwards, the gel pieces were distained in 200 mM ammonium bicarbonate/50% acetonitrile for 15 min, followed by 5 min in 100% acetonitrile. Proteins were reduced by adding 20 mM dithiothreitol in 25 mM ammonium bicarbonate and incubated for 20 min at 55 °C. The mixture was cooled to room temperature, and then, free thiols were alkylated by adding 40 mM iodoacetamide in 25 mM ammonium bicarbonate for 20 min in the dark. Finally, the gel pieces were washed twice in 25 mM ammonium bicarbonate.

Proteolytic digestion was performed by adding trypsin to a final concentration of 12.5 ng/µL in 25 mM ammonium bicarbonate at 37 °C overnight. Protein digestion was stopped by adding trifluoroacetic acid at a final concentration of 1% (*v*/*v*). Finally, the digested samples were vacuum-dried and dissolved in a mixture of 2% (*v*/*v*) acetonitrile and 0.05% (*v*/*v*) trifluoroacetic acid.

### 4.3. Shotgun-DDA-LC-MS/MS Analysis

Peptide separations were performed on a nano-LC using Dionex Ultimate 3000 nano UPLC (Thermo Scientific, San Jose, CA, USA), equipped with a C18 75 μm × 50 cm Acclaim Pepmap column (Thermo Scientific, San Jose, CA, USA), at 40 °C, at a flow rate of 300 nL/min. Peptide mixtures were previously concentrated and cleaned on a 300 µm × 5 mm Acclaim Pepmap precolumn (Thermo Scientific, San Jose, CA, USA) using 2% acetonitrile/0.05% trifluoroacetic acid, at 5 µL/min, for 5 min. Peptides were eluted with a gradient of 60 min ranging from 96% solvent A (0.1% formic acid) to 90% solvent B (80% acetonitrile and 0.1% formic acid), followed by an 8 min wash at 90% solvent B and a 12 min re-equilibration at 4% solvent B. Eluted peptides were converted into gas-phase ions via nanoelectrospray ionization and analyzed on a Thermo Orbitrap Fusion mass spectrometer (Thermo Scientific, San Jose, CA, USA) operated in positive mode. Survey scans of peptide precursors were acquired over an m/z range 400–1500 at 120K resolution (at 200 *m*/*z*) with a 4 × 10^5^ ion count target. Tandem MS was performed via isolation at 1.2 Da with the quadrupole. Monoisotopic precursor ions were fragmented via CID (Chemically Induced Dimerization) in an ion trap, which was set up as follows: automatic gain control, 2 × 10^3^; maximum injection time, 50 ms; and normalized collision energy, 35%. Only those precursors with charge states of 2–5 were sampled for MS2. A dynamic exclusion time of 15 s and a tolerance of 10 ppm around the selected precursor and its isotopes were used to avoid redundant fragmentations. The instrument was run in top 30 mode with 3 s cycles, meaning the instrument would continuously perform MS2 events until a maximum of 30 non-excluded precursors or 3 s, whichever was shorter.

### 4.4. Protein Quantification

Charge state deconvolution and deisotoping were not performed. MS2 spectra were searched using MaxQuant software v. 1.5.7.4 [64]. MS2 spectra were searched with Andromeda engines against a database of Uniprot *Zea mays*_Jun19. Peptides generated from tryptic digestion were searched by employing the following parameters: up to one missed cleavage, the carbamidomethylation of cysteines as fixed modifications, and the oxidation of methionine as variable modifications. The precursor mass tolerance was 10 ppm and product ions were searched at 0.6 Da tolerances. A target-decoy search strategy was applied, which integrates multiple peptide parameters such as length, charge, number of modifications, and identification score into a single quality that acts as statistical evidence of the quality of each single peptide spectrum match. The identified peptides were grouped into proteins according to the law of parsimony and filtered to a 1% false discovery rate (FDR). Peptide quantification was carried out using MaxQuant software, via a MaxLFQ label-free quantification method [65]. In the MaxLFQ label-free quantification method, a retention time alignment and identification transfer protocol (“match-between runs” feature inMaxQuant) was applied. Proteins identified from only one peptide were not taken into account in this analysis. Peak intensities across the whole set of quantitative data for all peptides in the samples were imported from the LFQ intensities of proteins from the MaxQuant analysis and normalized according to Cox et al. [65]. LFQ-normalized intensity values were transformed to a logarithmic scale with a base of two. Protein quantification and the calculation of statistical significance were carried out using a Student‘s *t*-test and error correction (*p*-value ≤ 0.05). The criteria used to consider a protein as differentially expressed were as follows: (a) the protein was consistently present in at least three biological replicates per condition; (b) it had statistically significant differences (Student‘s *t*-test, *p* ≤ 0.05) between genotypes or B treatments; and (c) it had a fold change ≥1.5 or ≤0.66667. The differentially accumulated proteins were manually categorized by function using different databases (Uniprot, https://www.uniprot.org/; Maize Genetics and Genomics, https://www.maizegdb.org/; ExplorEnz, https://www.enzyme-database.org/; BRENDA, https://www.brenda-enzymes.org/; KEGG: Kyoto Encyclopedia of Genes and Genomes, https://www.genome.jp/kegg/; and PANTHER: Protein ANalysis THrough Evolutionary Relationships, http://pantherdb.org/, accessed between 6 June 2022 and 24 January 2023).

## 5. Conclusions

The higher B content in Pachía leaves than in Sama leaves can cause greater damage to their proteins. The overexpression of several proteases, mainly cysteine protease14 and four serine proteases, can increase the degradation of damaged and misfolded proteins in Pachía. Subsequently, the over-accumulation of transcription- and translation-related proteins allows Pachía to: (1) partially replace proteins damaged by B toxicity, and (2) reduce the injury caused to ribosomes and protein synthesis by B excess by abolishing the non-viable decrease in transcription and translation processes, thus allowing Pachía to survive under this stress condition.

In Sama, PAO1 over-accumulation can protect the photosynthetic apparatus. Furthermore, ZmPIFI and NDHS up-accumulation, along with the lower knockdown of several subunits of NDH and ATP synthase complexes under B excess, confers greater B toxicity tolerance to this landrace by: (1) acting as an electron safety valve that prevents stromal overreduction, and thus, decreases photosynthetic damage, and (2) providing an additional supply of ATP that contributes to repairing the photosynthetic system of Sama. On the other hand, OEE2-1 overexpression can stabilize and protect the OEC, leading to a higher photosynthesis rate in this landrace.

## Figures and Tables

**Figure 1 plants-12-02322-f001:**
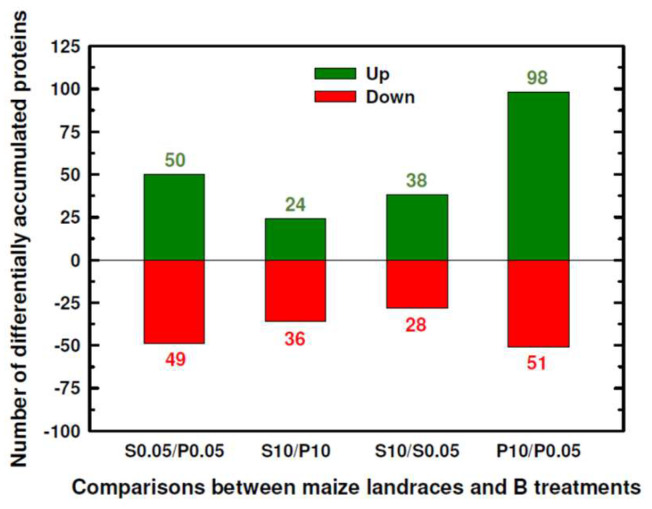
Number of significantly (*p* ≤ 0.05) up- or down-accumulated proteins, represented as positive and negative, respectively, compared to maize landraces and B treatments. Seedlings were subjected to 0.05 and 10 mM B for 10 days. Results were obtained from 3–4 separate plants of each landrace and B treatment. For more details, see Section 4. S: Sama landrace; P: Pachía landrace; 0.05: 0.05 mM B (B control treatment); 10: 10 mM B (B toxicity treatment). The numbers above the columns represent the numbers of proteins accumulated up (green) or down (red).

**Figure 2 plants-12-02322-f002:**
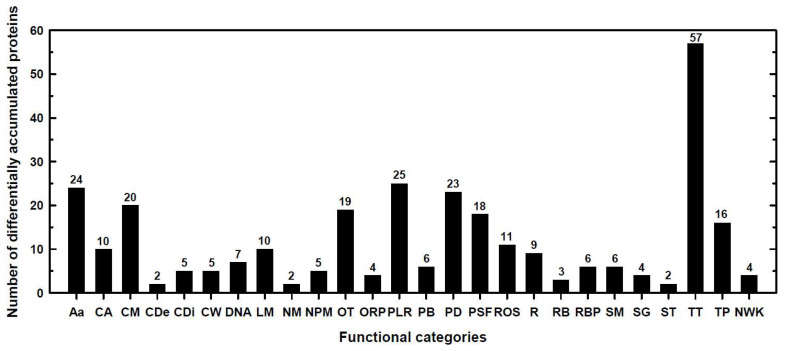
Number of differentially accumulated proteins (DAPs) in the different functional categories obtained from the four comparisons shown in Figure 1 and Appendix A. Seedlings of Sama and Pachía landraces were subjected to 0.05 and 10 mM B for 10 days. Results were obtained through the addition of the DAPs in the four comparisons. For more details, see Section 4. Aa: amino acid metabolism; CA: carbon assimilation and Calvin cycle; CM: carbohydrate metabolism; CDe: cell death; CDi: cell division; CW: cell wall; DNA: DNA and chromatin organization and DNA repair; LM: lipid metabolism; NM: nitrogen metabolism; NPM: nucleotide, purine, and pyrimidine metabolism; OT: others; ORP: oxidation and reduction processes; PLR: photosynthetic light reactions; PB: pigment biosynthesis; PD: protein degradation; PSF: protein stabilization and folding; ROS: reactive oxygen species scavenging pathways/response to oxidative stress; R: respiration metabolism (glycolysis, TCA cycle, and mitochondrial electron transfer); RB: ribosome biogenesis; RBP: RNA binding and processing; SM: secondary metabolism; SG: signaling; ST: stress; TT: transcription and translation processes; TP: transporters and transport processes; NWK: not well-known proteins.

**Figure 3 plants-12-02322-f003:**
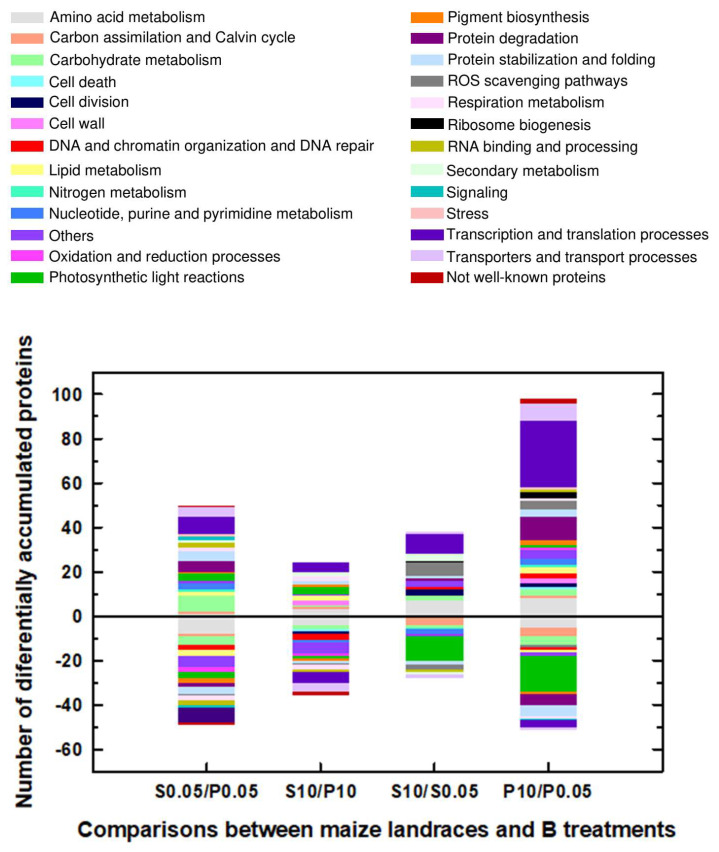
Functional categories of 303 maize proteins given as the number of those significantly expressed, represented as positive (up-accumulated) and negative (down-accumulated). Seedlings of Sama (S) and Pachía (P) landraces were subjected to 0.05 and 10 mM B for 10 days. Results were obtained from 3-4 separate plants of each landrace and B treatments. For more details, see Section 4. 0.05: 0.05 mM B (B control treatment); 10: 10 mM B (B toxicity treatment).

**Figure 4 plants-12-02322-f004:**
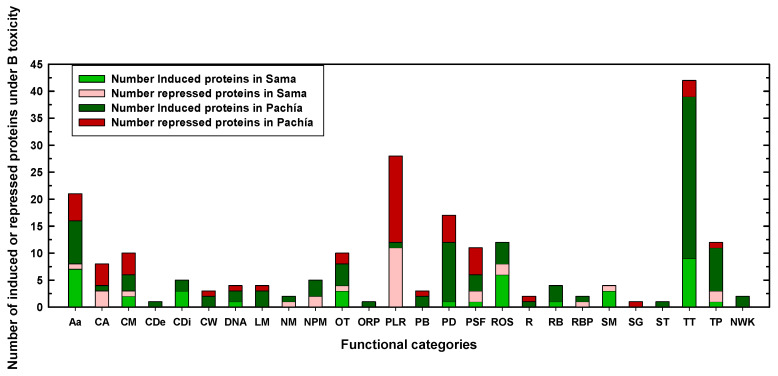
Number of induced or repressed proteins in Sama and Pachía landraces in the different functional categories, obtained from the comparisons between B toxicity and B control conditions shown in Appendix A. Seedlings of Sama and Pachía landraces were subjected to 0.05 (control) and 10 mM (toxicity) B for 10 days. Results were obtained through addition of induced or repressed proteins in Sama and Pachía. For more details, see Section 4. Aa: amino acid metabolism; CA: carbon assimilation and Calvin cycle; CM: carbohydrate metabolism; CDe: cell death; CDi: cell division; CW: cell wall; DNA: DNA and chromatin organization and DNA repair; LM: lipid metabolism; NM: nitrogen metabolism; NPM: nucleotide, purine, and pyrimidine metabolism; OT: others; ORP: oxidation and reduction processes; PLR: photosynthetic light reactions; PB: pigment biosynthesis; PD: protein degradation; PSF: protein stabilization and folding; ROS: reactive oxygen species scavenging pathways/response to oxidative stress; R: respiration metabolism (glycolysis, TCA cycle, and mitochondrial electron transfer); RB: ribosome biogenesis; RBP: RNA binding and processing; SM: secondary metabolism; SG: signaling; ST: stress; TT: transcription and translation processes; TP: transporters and transport processes; NWK: not well-known proteins.

**Table 1 plants-12-02322-t001:** Number of proteins detected in leaves of Pachía (P) and Sama (S) landraces under different boron (B) treatments and number of significantly differentially accumulated proteins (DAPs) in Pachía and Sama landraces under different B treatments.

	P0.05 mM (Control)	P10 mM B (B Toxicity)	S0.05 mM (Control)	S10 mM (B Toxicity)
Number of detected proteins ^1^	1100	1040	1111	1145
		S0.05 versus P0.05 (control conditions)	S10 versus P10 (B toxicity conditions)	
Number of significant DAPs between Sama and Pachía		99	60	
		Sama S10 versus S0.05	Pachía P10 versus P0.05	
Number of significant DAPs by B toxicity		66	149	

^1^ Numbers of proteins that were detected in at least one landrace (Sama or Pachía) and one B treatment analyzed.

**Table 2 plants-12-02322-t002:** Commonly expressed proteins in both Pachía and Sama landraces in response to boron (B) toxicity.

			Pachía		Sama			
Protein ID ^1^	Gene Name/ID ^2^	Protein Name/ Annotation	FC ^3^	*p*-Value ^4^	FC ^3^	*p*-Value ^4^	FCSA/ FCPA ^5^	Function/Biological Process ^6^
**AMINO ACID METABOLISM**								
B6SKB7	Zm00001d031013	Methylcrotonoyl-CoA carboxylase subunit α	4.44	0.0022	3.56	0.0049	0.80	Leucine degradation
A0A1D6K836	Zm00001d029848	Branched-chain amino-acid aminotransferase	2.35	0.0272	1.65	0.0241	0.70	Branched-chain amino acid biosynthesis
B4G011	Zm00001d046923	d-3-phosphoglycerate dehydrogenase chloroplastic	2.31	0.0154	1.52	0.0202	0.66	Serine biosynthesis
A0A1D6DW07	Zm00001d002051	d-3-phosphoglycerate dehydrogenase	1.78	0.0494	1.69	0.0175	0.95	Serine biosynthesis
**CARBON ASSIMILATION/CALVIN CYCLE**								
O24574	Zm00001d004894	Ribulose bisphosphate carboxylase small chain	0.38	0.0113	0.33	0.0466	0.87	Carbon dioxide fixation
**CARBOHYDRATE METABOLISM**								
Q9FQ11	Zm00001d010523	Sucrose-phosphatase 1	1.50	0.0154	1.58	0.0420	1.05	Sucrose biosynthesis
A0A1D6IJ76	Zm00001d022107	Glyceraldehyde-3-phosphate dehydrogenase A	0.34	0.0319	0.51	0.0019	1.52	Carbon metabolism
**CELL DIVISION**								
A0A1D6FRI4	Zm00001d010500	ERBB-3 binding protein 1	1.89	0.0387	1.58	0.0266	0.84	Cell division and cell growth regulation
**PHOTOSYNTHETIC LIGHT REACTIONS**								
A0A1D6HS38	Zm00001d018779	Oxygen-evolving enhancer protein 2-1 chloroplastic (OEE2-1)	0.27	0.0110	0.48	0.0354	1.78	Photosynthesis. Photosystem II oxygen-evolving complex
B4FWG2	Zm00001d048422	Photosynthetic NDH subunit of subcomplex B 2 chloroplastic	0.25	0.0047	0.41	0.0200	1.62	Photosynthetic electron transport flow around photosystem I to produce ATP
A0A1X7YHG9	AtpA	ATP synthase subunit α chloroplastic (ATPα)	0.20	0.0166	0.61	0.0163	2.99	Chloroplast ATP synthesis-coupled proton transport
P46617	PetA	Cytochrome f	0.18	0.0193	0.29	0.0161	1.59	Photosynthetic electron transport activity
P00827	Zm00001d006403	ATP synthase subunit β chloroplastic (ATPβ)	0.15	0.0076	0.52	0.0274	3.45	Chloroplast ATP synthesis-coupled proton transport
**REACTIVE OXYGEN SPECIES (ROS) SCAVENGING PATHWAYS/RESPONSE TO OXIDATIVE STRESS**								
A0A1D6MSE3	Zm00001d040721	Dihydrolipoyl dehydrogenase	2.30	0.0273	1.80	0.0205	0.78	Cell redox homeostasis
A0A1D6JPH3	Zm00001d027769	Glutathione reductase	2.21	0.0053	1.71	0.0436	0.77	Cell redox homeostasis. Glutathione metabolic process. Cellular oxidant detoxification
**RIBOSOME BIOGENESIS**								
K7UTH7	Zm00001d009596	GTPase ERA1 chloroplastic	2.61	0.0108	1.81	0.0126	0.69	Ribosome biogenesis. Ribosomal small subunit assembly. rRNA processing
**TRANSCRIPTION AND TRANSLATION PROCESSES**								
A0A1D6LIV5	Zm00001d035802	Phenylalanine–tRNA ligase beta subunit cytoplasmic	2.56	0.0314	2.23	0.0093	0.87	Translation. Phenylalanyl-tRNA aminoacylation
**TRANSPORTERS AND TRANSPORT PROCESSES**								
B6SP43	Zm00001d007597	ABC family1	4.54	0.0103	2.69	0.0125	0.59	ATPase-coupled transmembrane transporter activity

^1^ Protein ID: protein identification (ID) number in the UniProt database; ^2^ Gene Name: name or ID number of the corresponding gene of the differentially expressed protein as searched in the Maize Genetics and Genomics Database (MaizeGDB; https://www.maizegdb.org/, accessed between 6 June 2022 and 24 January 2023). ^3^ Fold Change is expressed as the ratio of LFQ intensities (on a logarithmic scale) of proteins between 10 and 0.05 mM B treatments; Induced proteins are highlighted with light green rows and repressed proteins with light red rows. ^4^
*p*-value: statistical level (using Student’s *t*-test) ≤ 0.05, at which differential protein expression was accepted as significant; ^5^ FCSA/FCPA, is the ratio between fold change of Sama and Pachía. ^6^ Function/Biological process: annotated biological functions or biological process based on different databases. Induced proteins are highlighted with light green rows and repressed proteins with light red rows. For more details, see Section 4. Results were obtained from 3–4 separate plants of each landrace.

**Table 3 plants-12-02322-t003:** Proteins with higher differential expression in Pachía leaves in response to boron (B) toxicity. This table shows the proteins that were very strongly induced or repressed by B toxicity in Pachía by comparing their expressions with those of Pachía in medium with 0.05 mM B.

Protein ID ^1^	Gene Name/ID ^2^	Protein Name/Annotation	FC ^3^	*p*-Value ^4^	Function/Biological Process ^5^
**AMINO ACID AND PEPTIDE METABOLISMS**					
**Proteins very strongly induced by B toxicity in Pachía**					
B6SKB7	Zm00001d031013	Methylcrotonoyl-CoA carboxylase subunit α	4.44	0.0022	Leucine degradation
**Proteins very strongly repressed by B toxicity in Pachía**					
A0A1D6ICL3	Zm00001d021596	Adenosine 5-phosphosulfate reductase-like1	0.29	0.0140	Cysteine biosynthetic process. Sulfate reduction
B6TZD1	Zm00001eb168430	Methylthioribose-1-phosphate isomerase	0.24	0.0461	Methionine biosynthesis
**CARBON ASSIMILATION AND CALVIN CYCLE**					
**Proteins very strongly repressed by B toxicity in Pachía**					
B4FQ59	Zm00001d017711	Phosphoribulokinase	0.33	0.0004	Calvin–Benson cycle
Q9ZT00	Zm00001eb164390	Ribulose bisphosphate carboxylase/oxygenase activase chloroplastic	0.26	0.0090	Carbon dioxide fixation. Rubisco activator activity
**CELL WALL**					
**Proteins very strongly induced by B toxicity in Pachía**					
B4F9J1	Zm00001d046357	β-galactosidase	3.17	0.0092	Xyloglucan degradation
**DNA AND CHROMATIN ORGANIZATION AND DNA REPAIR**					
**Proteins very strongly induced by B toxicity in Pachía**					
B6TGH8	Zm00001d034479	Histone H1	3.60	0.0349	Chromosome condensation. Nucleosome assembly. Nucleosome positioning
C0P6Q6	Zm00001d040416	DNA gyrase subunit B	3.48	0.0007	DNA topological change
**LIPID METABOLISM**					
**Proteins very strongly repressed by B toxicity**					
B4FLS8	Zm00001d003584	12-oxo-phytodienoic acid reductase 5	0.33	0.0436	Fatty acid and oxylipin biosynthesis
**OTHERS**					
**Proteins very strongly repressed by B toxicity in Pachía**					
C0PE12	Zm00001d009877	Protein plastid transcriptionally active 16 chloroplastic	0.24	0.0121	Circadian rhythm
**PHOTOSYNTHETIC LIGHT REACTIONS**					
**Proteins very strongly repressed by B toxicity in Pachía**					
B6SP99	Zm00001d024148	Photosynthetic NDH subunit of subcomplex B 1 chloroplastic	0.33	0.0137	Photosynthetic electron transport in photosystem I
B4FJP7	Zm00001d027729	Photosynthetic NDH subunit of subcomplex B 2 chloroplastic	0.32	0.0169	Photosynthetic electron transport in photosystem I
B4FR80	Zm00001d033098	Post-illumination chlorophyll fluorescence increase (ZmPIFI)	0.28	0.0270	Chlororespiration
A0A1D6HS38	Zm00001d018779	Oxygen-evolving enhancer protein 2-1 chloroplastic (OEE2-1)	0.27	0.0110	Photosynthesis. Photosystem II oxygen-evolving complex
B4FWG2	Zm00001d048422	Photosynthetic NDH subunit of subcomplex B 2 chloroplastic	0.25	0.0047	Photosynthetic electron transport flow around photosystem I to produce ATP
P19124	NdhJ	NAD(P)H-quinone oxidoreductase subunit J, chloroplastic	0.22	0.0147	Photosynthesis, light reaction, photosynthetic electron transport chain. Couples the photosynthetic redox reaction to proton translocation
A0A1X7YHG9	AtpA	ATP synthase subunit α (ATPα)	0.20	0.0166	Chloroplast ATP synthesis-coupled proton transport
P46617	PetA	Cytochrome f	0.18	0.0193	Photosynthetic electron transport chain
P00827	Zm00001d009488	ATP synthase subunit β, chloroplastic (ATPβ)	0.15	0.0076	Chloroplast ATP synthesis-coupled proton transport
A0A1D6JYG6	Zm00001d028670	Photosynthetic NDH subunit of lumenal location 1 chloroplastic	0.13	0.0134	Part of photosystem II oxygen-evolving complex
**PROTEIN DEGRADATION**					
**Proteins very strongly induced by B toxicity in Pachía**					
B4FS65	Zm00001d005391	Cysteine protease 14	4.38	0.0146	Proteolysis. Proteolysis involved in protein catabolic process
A0A1D6HM49	Zm00001d018282	Subtilisin-like protease SBT1.4	3.70	0.0399	Serine protease. Serine-type endopeptidase activity. Proteolysis
A0A1D6H4R4	Zm00001d015962	Prolyl oligopeptidase family protein	3.58	0.0080	Proteolysis. Serine protease. Serine-type peptidase activity
**PROTEIN STABILIZATION AND FOLDING**					
**Proteins very strongly repressed by B toxicity in Pachía**					
G2XK63	Zm00001d040257	T-complex protein 1 subunit β	0.27	0.0065	Protein folding. Chaperone
B4FR04	Zm00001d019052	Peptidylprolyl isomerase	0.23	0.0205	Protein folding. Rotamase
**SIGNALING**					
**Proteins very strongly repressed by B toxicity in Pachía**					
P49235	Zm00001eb411380	4-hydroxy-7-methoxy-3-oxo-3,4-dihydro-2*H*-1,4-benzoxazin-2-yl glucoside beta-d-glucosidase 1, chloroplastic	0.19	0.0090	Cytokinin signaling pathway
**TRANSCRIPTION AND TRANSLATION PROCESSES**					
**Proteins very strongly induced by B toxicity in Pachía**					
A0A1D6LEN8	Zm00001d035139	MA3 domain-containing protein	4.95	0.0073	Negative regulation of transcription, DNA-templated. Regulation of translation
Q6R9D1	GRMZM5G806488	Ribosomal protein S7	3.89	0.0202	Translation. Ribosomal small subunit assembly. Structural constituent of ribosomes
A0A1D6IAN8	Zm00001d021400	Octicosapeptide/Phox/Bem1p (PB1) domain-containing protein/tetratricopeptide repeat (TPR)-containing protein	3.47	0.0323	RNA processing
C0P456	Zm00001d002789	Pentatricopeptide repeat-containing protein	3.26	0.0259	Likely involved in post-transcriptional control of gene expression in organelles
**Proteins very strongly repressed by B toxicity in Pachía**					
O50018	Zm00001d046449	Elongation factor 1-α	0.29	0.0269	Translation. Translation elongation factor activity
**TRANSPORTERS AND TRANSPORT PROCESSES**					
**Proteins very strongly induced by B toxicity in Pachía**					
B6SP43	Zm00001d007597	ABC family1	4.54	0.0103	ATPase-coupled transmembrane transporter activity
A0A1D6H2R4	Zm00001d015569	H^+^-exporting diphosphatase	4.34	0.0050	Ion transport. Pyrophosphate hydrolysis-driven proton transmembrane transporter activity
A0A1D6MS70	Zm00001d040686	Protein translocase subunit SECA1 chloroplastic	4.12	0.0173	Protein transport
A0A1D6DSW6	Zm00001d001788	K^+^ efflux antiporter 2 chloroplastic	3.79	0.0414	Chloroplast potassium ion trans-port
B6T5R1	Zm00001d010504	Ran-binding protein 1	3.16	0.0492	Intracellular transport. Protein and mRNA transport. Nucleocytoplasmic transport

Only proteins considered differentially expressed, namely those with fold changes ≥3.0 or ≤0.334 and *p*-values ≤ 0.05, are shown in this table. Very strongly induced proteins are highlighted with light green rows and very strongly repressed proteins with light red rows. ^1^ Protein ID: protein identification number in the UniProt database. ^2^ Gene Name: name or ID number of the corresponding gene of the differentially expressed protein as searched in the Maize Genetics and Genomics Database (MaizeGDB; https://www.maizegdb.org/, accessed between 6 June 2022 and 24 January 2023). ^3^ Fold Change is expressed as the ratio of LFQ intensities (on a logarithmic scale) of proteins between 10 and 0.05 mM B treatments in Pachía. Results were obtained from 3–4 separate plants. ^4^
*p*-value: statistical level (using Student’s *t*-test) below ≤0.05 at which differential protein expression was accepted as significant. ^5^ Function/Biological process: annotated biological functions or biological process based on different databases. For more details, see Section 4.

**Table 4 plants-12-02322-t004:** Proteins with higher differential expression in Sama leaves in response to boron (B) toxicity. This table shows the proteins that are very strongly induced or repressed by B toxicity in Sama by comparing their expressions with those of Sama in medium with 0.05 mM B.

Protein ID ^1^	Gene Name/ID ^2^	Protein Name/Annotation	FC ^3^	*p*-Value ^4^	Function/Biological Process ^5^
**AMINO ACID AND PEPTIDE METABOLISMS**					
**Proteins very strongly induced by B toxicity in Sama**					
B6SKB7	Zm00001d031013	Methylcrotonoyl-CoA carboxylase subunit alpha	3.56	0.0049	Leucine degradation
**CARBON ASSIMILATION AND CALVIN CYCLE**					
**Proteins very strongly repressed by B toxicity in Sama**					
O24574	Zm00001d004894	Ribulose bisphosphate carboxylase small chain	0.33	0.0466	Carbon dioxide fixation
P05348	Rbcs	Ribulose bisphosphate carboxylase small chain, chloroplastic	0.13	0.0096	Carbon dioxide fixation
**CELL DIVISION**					
**Proteins very strongly induced by B toxicity in Sama**					
C0P4T2	Zm00001d042664	Patellin-1	3.05	0.0149	Cell division and cell cycle
**PHOTOSYNTHETIC LIGHT REACTIONS**					
**Proteins very strongly repressed by B toxicity in Sama**					
P46617	PetA	Cytochrome f	0.29	0.0161	Photosynthetic electron transport chain
B6SQV5	Zm00001d049387	Photosystem II 10 kDa polypeptide	0.14	0.0438	Photosynthesis. Photosystem II oxygen-evolving complex
**PROTEIN STABILIZATION AND FOLDING**					
**Proteins very strongly repressed by B toxicity in Sama**					
C4J6Y2	Zm00001d018077	Peptidylprolyl isomerase	0.18	0.0422	Protein folding. Rotamase
**REACTIVE OXYGEN SPECIES (ROS) SCAVENGING PATHWAYS/RESPONSE TO OXIDATIVE STRESS**					
**Proteins very strongly repressed by B toxicity in Sama**					
B4FZ35	Zm00001d002240	CHL-*Zea mays* chloroplastic lipocalin	0.31	0.0272	Response to oxidative stress. Violaxanthin, antheraxanthin, and zeaxanthin interconversion
**SECONDARY METABOLISM**					
**Proteins very strongly induced by B toxicity in Sama**					
O64411	Zm00001d024281	Polyamine oxidase 1 (PAO1)	3.34	0.0108	Spermine degradation. Amine and polyamine degradation

Only proteins considered differentially expressed, namely those with fold-changes ≥3.0 or ≤0.334 and *p*-values ≤ 0.05, are shown in this table. Very strongly induced proteins are highlighted with light green rows and very strongly repressed proteins with light red rows. ^1^ Protein ID: protein identification number in the UniProt database. ^2^ Gene Name: name or ID number of the corresponding gene of the differentially expressed protein as searched in the Maize Genetics and Genomics Database (MaizeGDB; https://www.maizegdb.org/, accessed between 6 June 2022 and 24 January 2023). ^3^ Fold Change is expressed as the ratio of LFQ intensities (on a logarithmic scale) of proteins between 10 and 0.05 mM B treatments in Sama. Results were obtained from 3–4 separate plants. ^4^
*p*-value: statistical level (using Student’s *t*-test), at which differential protein expression was accepted as significant (≤0.05). ^5^ Function/Biological process: annotated biological functions or biological process based on different databases. For more details, see Section 4.

**Table 5 plants-12-02322-t005:** Proteins with higher differential expression between Sama and Pachía leaves under boron (B) toxicity conditions. This table shows the strongly up- or down-accumulated proteins in Sama in media with 10 mM B compared to those of Pachía in 10 mM B.

Protein ID ^1^	Gene Name/ID ^2^	Protein Name/Annotation	FC ^3^	*p*-Value ^4^	Function/Biological Process ^5^
**AMINO ACID AND PEPTIDE METABOLISMS**					
**Strongly up-accumulated proteins in Sama in media with 10 mM B**					
A0A1D6ICL3	Zm00001d021596	Adenosine 5-phosphosulfate reductase-like1	2.33	0.0417	Cysteine biosynthetic process. Sulfate reduction
**CARBON ASSIMILATION AND CALVIN CYCLE**					
**Strongly up-accumulated proteins in Sama in media with 10 mM B**					
A0A1D6EXF1	Zm00001d006520	PDK regulatory protein1	2.16	0.0167	Regulation of C4 photosynthetic carbon assimilation cycle
**CARBOHYDRATE METABOLISM**					
**Strongly up-accumulated proteins in Sama in media with 10 mM B**					
Q9SYS1	Zm00001d021702	β-amylase	2.63	0.0499	β-amylase activity. Starch degradation
**Strongly down-accumulated proteins in Sama in media with 10 mM B**					
A0A1D6K5L6	Zm00001d029502	Glucose-6-phosphate 1-dehydrogenase	0.36	0.0411	Pentose phosphate pathway
A0A1D6LY56	Zm00001d037480	Alkaline α galactosidase 2	0.33	0.0438	Carbohydrate metabolic process
**CELL DEATH**					
**Strongly down-accumulated proteins in Sama in media with 10 mM B**					
A0A1D6JNJ8	Zm00001d027656	Lethal leaf-spot 1	0.32	0.0016	Cell death. Chlorophyll catabolic process
**CELL DIVISION**					
**Strongly down-accumulated proteins in Sama in media with 10 mM B**					
A0A1D6JH24	Zm00001d026532	Protein RCC2	0.42	0.0214	Cell division
**CELL WALL**					
**Strongly up-accumulated proteins in Sama in media with 10 mM B**					
A0A1D6MWZ7	Zm00001d041578	Glossy6	3.27	0.0403	Epicuticular wax accumulation. Intracellular trafficking of cuticular waxes
**DNA AND CHROMATIN ORGANIZATION AND DNA REPAIR**					
**Strongly down-accumulated proteins in Sama in media with 10 mM B**					
B4FQA5	Zm00001d018981	Histone1a	0.35	0.0318	Chromosome condensation. Nucleosome assembly
B6TGH8	Zm00001d034479	Histone H1	0.31	0.0138	Chromosome condensation. Nucleosome assembly. Nucleosome positioning
**LIPID METABOLISM**					
**Strongly up-accumulated proteins in Sama in media with 10 mM B**					
Q8W0V2	Zm00001d033623	Lipoxygenase 3	5.06	0.0455	Fatty acid and oxylipin biosynthesis
Q06XS3	Zm00001d053675	Lipoxygenase 10	3.44	0.0247	Fatty acid and oxylipin biosynthesis
**OTHERS**					
**Strongly down-accumulated proteins in Sama in media with 10 mM B**					
B6TY16	Zm00001d040331	SUN domain protein2	0.41	0.0262	Nuclear envelope organization
B4F7V3	Zm00001d021582	Protein phosphatase 2C isoform ε	0.39	0.0214	Protein dephosphorylation
A0A1D6HUN3	Zm00001d019040	D-2-hydroxyglutarate dehydrogenase mitochondrial	0.33	0.0024	Photorespiration
**OXIDATION AND REDUCTION PROCESSES**					
**Strongly down-accumulated proteins in Sama in media with 10 mM B**					
B4F987	Zm00001d020984	Putative sarcosine oxidase	0.23	0.0321	Sarcosine oxidase activity
**PHOTOSYNTHETIC LIGHT REACTIONS**					
**Strongly up-accumulated proteins in Sama in media with 10 mM B**					
B4FR80	Zm00001d033098	Post-illumination chlorophyll fluorescence increase (ZmPIFI)	2.52	0.0097	Chlororespiration
A0A1D6HS38	Zm00001d018779	Oxygen-evolving enhancer protein 2-1 chloroplastic (OEE2-1)	2.31	0.0325	Photosynthesis. Photosystem II oxygen-evolving complex
**PIGMENT BIOSYNTHESIS**					
**Strongly up-accumulated proteins in Sama in media with 10 mM B**					
A0A1D6JHX0	Zm00001d026603	Magnesium-chelatase subunit ChlH1 chloroplastic (ChlH1)	2.90	0.0484	Chlorophyll biosynthetic process
**PROTEIN STABILIZATION AND FOLDING**					
**Strongly down-accumulated proteins in Sama in media with 10 mM B**					
A0A1D6KE29	Zm00001d030725	Heat shock protein 70	0.43	0.0406	Protein refolding. Protein folding chaperone. Cellular response to unfolded protein
**RESPIRATION (GLYCOLISIS, TCA CYCLE AND MITOCHONDRIAL ELECTRON TRANSFER)**					
**Strongly up-accumulated proteins in Sama in media with 10 mM B**					
B4G1C9	Zm00001d023606	Dihydrolipoamide acetyltransferase component of pyruvate dehydrogenase complex	2.04	0.0332	Acetyl-CoA biosynthetic process from pyruvate
**Strongly down-accumulated proteins in Sama in media with 10 mM B**					
A0A1D6MAK9	Zm00001d038792	Phosphotransferase	0.49	0.0331	Glycolysis
**SECONDARY METABOLISM**					
**Strongly up-accumulated proteins in Sama in media with 10 mM B**					
O64411	Zm00001d024281	Polyamine oxidase 1 (PAO1)	5.15	0.0007	Spermine degradation. Amine and polyamine degradation
**TRANSCRIPTION AND TRANSLATION**					
**Strongly up-accumulated proteins in Sama in media with 10 mM B**					
B4FP25	Zm00001d047296	40S ribosomal protein S19	6.38	0.0289	Translation. Structural constituent of ribosome. Ribosomal small subunit assembly
B6TDF7	Zm00001d019898	Plastid-specific 30S ribosomal protein 2	2.31	0.0243	Ribosomal protein. Ribonucleoprotein complex. RNA-binding
C0PEC4	Zm00001d032420	30S ribosomal protein S5 chloroplastic	2.12	0.0487	Translation. Structural constituent of ribosome
**Strongly down-accumulated proteins in Sama in media with 10 mM B**					
B6SX73	Zm00001d016549	60S ribosomal protein L35	0.42	0.0284	Translation. Structural constituent of ribosome
Q6R9D1	GRMZM5G806488	Ribosomal protein S7	0.35	0.0426	Translation. Structural constituent of ribosome. Ribosomal small subunit assembly
**TRANSPORTER AND TRANSPORT PROCESSES**					
**Strongly down-accumulated proteins in Sama in media with 10 mM B**					
A0A1D6H2R4	Zm00001d015569	H^+^-exporting diphosphatase	0.33	0.0169	Ion transport. Pyrophosphate hydrolysis-driven proton transmembrane transporter activity
A0A1D6K7N5	Zm00001d029762	Hexose transporter	0.20	0.0439	Hexose transporter
NOT WELL-KNOWN PROTEINS					
**Strongly down-accumulated proteins in Sama in media with 10 mM B**					
A0A1D6KKK1	Zm00001d031677	MtN19-like protein	0.23	0.0121	Not well determined

Only proteins considered differentially expressed, namely those with fold-changes ≥2.0 or ≤0.5 and *p*-values ≤ 0.05, are shown in this table. Induced proteins are highlighted with light green rows and repressed proteins with light red rows. ^1^ Protein ID: protein identification number in the UniProt database. ^2^ Gene Name: name or ID number of the corresponding gene of the differentially expressed protein as searched in the Maize Genetics and Genomics Database (MaizeGDB; https://www.maizegdb.org/, accessed between 6 June 2022 and 24 January 2023). ^3^ Fold Change is expressed as the ratio of LFQ intensities (on a logarithmic scale) of proteins between Sama and Pachía in media with 10 mM B. Results were obtained from 3–4 separate plants. ^4^
*p*-value: statistical level (using Student’s *t*-test) below ≤0.05 at which protein differential expression was accepted as significant. ^5^ Function/Biological process: annotated biological functions or biological process based on different databases. For more details, see Section 4.

**Table 6 plants-12-02322-t006:** Proteins exclusively detected in Pachía or Sama leaves in at least one B treatment.

Protein ID ^1^	Gene Name/ID ^2^	Protein Name/Annotation	Function/Biological Process ^3^
**DNA AND CHROMATIN ORGANIZATION AND DNA REPAIR**			
**Proteins exclusively detected in Pachía in 10 mM B**			
A0A1D6KX75	Zm00001d033247	Nfc103a–nucleosome/chromatin assembly factor C	Nucleosome/chromatin assembly. DNA repair. Chromatin remodeling, regulation of DNA-templated transcription
**OTHERS**			
**Proteins exclusively detected in Sama in both B treatments**			
K7VAT7	Zm00001d046569	Protein kinase superfamily protein with octicosapeptide/Phox/Bem1p domain	Protein serine/threonine kinase activity. Protein phosphorylation
**REACTIVE OXYGEN SPECIES (ROS) SCAVENGING PATHWAYS/RESPONSE TO OXIDATIVE STRESS**			
**Proteins exclusively detected in Pachía in both B treatments**			
B4FKV6	Zm00001d014341	Peroxidase 54	Response to oxidative stress. Peroxidase activity
**TRANSCRIPTION AND TRANSLATION**			
**Proteins exclusively detected in Pachía in 10 mM B**			
A0A096RFR6	Zm00001d039518	Eukaryotic translation initiation factor 3 subunit A (eIF3a)	Translation initiation factor activity. Protein synthesis. Formation of cytoplasmic translation initiation complex
**TRANSPORTER AND TRANSPORT PROCESSES**			
**Proteins exclusively detected in Pachía in both B treatments**			
A0A1D6EU13	Zm00001d006238	Calcium lipid binding protein-like	Lipid transport
A0A1D6JN64	Zm00001d027580	Outer mitochondrial membrane porin1 (ommp1)	Voltage-gated anion channel activity. Inorganic anion transport, transmembrane transport, anion transmembrane transport
**Proteins exclusively detected in Sama in both B treatments**			
Q7Y1W6	Zm00001d018693	Pentatricopeptide repeat 2 (PPR2)	Chloroplast translation
NOT WELL-KNOWN PROTEINS			
**Proteins exclusively detected in Sama in both B treatments**			
A0A1D6DWG9	Zm00001d002089	Tetratricopeptide repeat (TPR)-like superfamily protein	Unknown

^1^ Protein ID: protein identification number in the UniProt database. ^2^ Gene Name: name or ID number of the corresponding gene of the identified protein as searched in the Maize Genetics and Genomics Database (MaizeGDB; https://www.maizegdb.org/, accessed between 6 June 2022 and 24 January 2023). ^3^ Function/Biological process: annotated biological functions or biological process based on different databases. For more details, see Section 4.

## Data Availability

The data presented in this study are available in the text and Appendix A.

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
