# Peer review of "Leaf Proteomic Analysis in Seedlings of Two Maize Landraces with Different Tolerance to Boron Toxicity"

_plants, 2023, doi:10.3390/plants12122322_

Round 1

Reviewer 1 Report

The manuscript is well written except for few Minor Revisions given below:

(1) Title: Since the evaluation was done in the seedling stage, it must be reflected in the title of the manuscript

(2) Abstract: Line No. 13: rewrite as 'quality of the produce'

(3) Introduction: There is no mention regarding why leaf samples are preferred over other tissues.

(4) Results: Figure 2: Plot the number of DAPs in the bar graph for easy understanding to the readers

(5) Results: Table 3 & Table 4 can be taken as Supplementary tables, as Table 5 and 6 conveys the significant outcome of the study

(6) Discussion: Kindly discuss about the applicability of these findings in the seedling stage in the vegetative, flowering and grain filling stage citing available literature

(7) Conclusion: Elaborate more on the significant findings of the study

Reviewer 2 Report

The work of Betty M. Mamani-Huarcaya on Leaf proteomic analysis in two maize landraces with different tolerance to boron toxicity. The results suggested that higher tolerance to B toxicity of Sama can be attributed to more stable photosynthesis that would avoid damage caused by stromal over-reduction under this stress condition.

Overall the work is interested and well written, however I thing the authors has stress more the novelty of the work and the main work messages. To achieve this, a summary model of the key findings will be helpful. The tables, particularly Table 3, is very long, I recommend to move the table to supplementary materials and only keep the key proteins as short table.

Reviewer 3 Report

The paper Leaf proteomic analysis in two maize landraces with different tolerance to boron toxicity written by Mamani-Huarcaya and coauthors presents the study results, that are dealing, in a broader sense, with a very interesting research area in ecology the variability of ecophysiological response in plants individuals to environmental and habitat stresses. It is a very interesting and increasingly developing area of research due to two main reasons. The unlimited abilities of plants and all living organisms to adapt particularly the ecophysiological ones and second the development of new analytic tools that enable scientists to track the mechanisms of adaptations performed by different organisms.

In the paper under review, the Authors have focused on Boron (B) toxicity as a challenging stress factor that negatively affects maize yield and the quality of maize plant individuals. The two Peruvian maize landraces, Sama and Pachía, were studied and physiologically characterized in terms of their tolerance to B toxicity. However, many aspects of the molecular mechanisms of these two maize landraces' resistance against B toxicity are still unknown.

The study is the more crucial as B is a micronutrient, the range between its deficient, optimal, and toxic levels for plants is very narrow. Moreover, several photosynthetic parameters, such as 60 CO2 assimilation (PN), photosynthetic electron transport rate (ETR), maximum biomass, chlorophyll fluorescence (Fv/Fm), and CO2 use efficiency can easily become decreased under B toxicity conditions.

The Authors perform, a leaf proteomic analysis of Sama and Pachía. The functional analysis indicated that many of these 20 proteins are involved in crucial molecular processes such as transcription and translation processes, amino acids metabolism, photosynthesis, carbohydrate metabolism, protein degradation, and protein stabilization and folding.

In the paper, the Authors have stated the overexpression of some proteases and transcription- and translation-related proteins that enable the Pachía maize landraces, to degrade and replace partially the proteins damaged by boron toxicity achieving survival under this challenging stress condition.

In Sama maize landraces, PAO1 overaccumulation and weaker limitation of several subunits of NDH and ATP synthase complexes under boron excess would confer a greater boron toxicity tolerance to this Sama type. These would act as a safety system that would reduce stromal over reduction, and thus limit photosynthetic damage and, it would provide an additional supply of ATP that would participate in the repair process of the photosynthetic system of Sama.

The general comment and suggestion are related to the fact that the very interesting results are true for the strictly controlled laboratory conditions. It is necessary to add an extended passage to the introduction and discussion section dealing with the analysis of the applicability of the obtained results in field conditions. How far the factors occurring in the field often synergy relation between the factors can modify the plant individuals molecular response in both Sama and Pachía mize in comparison to those identified in the laboratory.

Reviewer 4 Report

Authors have studied leaf proteomic analysis in maize landrace with different tolerance to boron toxicity in detail. The topic is sound. The manuscript is well written. All sections are in detail, everything is clear and connecting, easy to follow. I dont have any comments or remarks. I want to congratulate the authors for the great work.

Author Response

We are very grateful to Reviewer 4 for his/her comments.